# Phenotypic delay in the evolution of bacterial antibiotic resistance: Mechanistic models and their implications

**Martín Carballo-Pacheco**[1], **Michael D. Nicholson**[1,2,3], **Elin E. Lilja**[1], **Rosalind J. Allen**[1,4], **Bartlomiej Waclaw**[1,4]*

**1** School of Physics and Astronomy, The University of Edinburgh, Edinburgh, United Kingdom, **2** Department of Data Sciences, Dana-Farber Cancer Institute, Boston, Massachusetts, United States of America, **3** Department of Biostatistics, Harvard T.H. Chan School of Public Health, Boston, Massachusetts, United States of America, **4** Centre for Synthetic and Systems Biology, The University of Edinburgh, Edinburgh, United Kingdom

☯ These authors contributed equally to this work.
* bwaclaw@staffmail.ed.ac.uk

## Abstract

Phenotypic delay—the time delay between genetic mutation and expression of the corresponding phenotype—is generally neglected in evolutionary models, yet recent work suggests that it may be more common than previously assumed. Here, we use computer simulations and theory to investigate the significance of phenotypic delay for the evolution of bacterial resistance to antibiotics. We consider three mechanisms which could potentially cause phenotypic delay: effective polyploidy, dilution of antibiotic-sensitive molecules and accumulation of resistance-enhancing molecules. We find that the accumulation of resistant molecules is relevant only within a narrow parameter range, but both the dilution of sensitive molecules and effective polyploidy can cause phenotypic delay over a wide range of parameters. We further investigate whether these mechanisms could affect population survival under drug treatment and thereby explain observed discrepancies in mutation rates estimated by Luria-Delbrück fluctuation tests. While the effective polyploidy mechanism does not affect population survival, the dilution of sensitive molecules leads both to decreased probability of survival under drug treatment and underestimation of mutation rates in fluctuation tests. The dilution mechanism also changes the shape of the Luria-Delbrück distribution of mutant numbers, and we show that this modified distribution provides an improved fit to previously published experimental data.

## Author summary

Understanding precisely how some bacteria survive exposure to antibiotics is a major research focus. Specific mutations in the bacterial genome are known to provide protection. However, it remains unclear how much time passes between a bacterium acquiring the genetic change and being able to tolerate antibiotics—termed the phenotypic delay—and what controls this delay. Here, using computer simulations and mathematical arguments we discuss three biologically plausible mechanisms of phenotypic delay. We

**Data Availability Statement:** All numerical values used to generate graphs are included in the Supplementary Information in the form of an Excel spreadsheet. The work uses published data of Ref.

[47]: Boe L, Tolker-Nielsen T, Eegholm KM, Spliid H, Vrang A. Fluctuation analysis of mutations to nalidixic acid resistance in Escherichia coli. J Bacteriol. 1994;176(10):2781–2787. doi:10.1128/jb.176.10.2781-2787.1994. The data is also included in the supplementary Excel table (included in this submission), S1 Table, sheet "Fig 5", panel b, column "Experimental data". All algorithms used to process the data are described in the manuscript.

**Funding:** RJA, MCP, and EL received funding from the European Research Council under Consolidator grant 682237 EVOSTRUC, https://erc.europa.eu/. BW received funding from the Royal Society of Edinburgh Personal Research Fellowship, https://www.rse.org.uk/. MN received funding from the EPSRC DTA PhD studentship, https://epsrc.ukri.org/. The funders had no role in the study design, data collection and analysis, decision to publish, or preparation of the manuscript.

**Competing interests:** The authors have declared that no competing interests exist.

investigate how each mechanism would affect the outcome of laboratory experiments often used to study the evolution of antibiotic resistance, and we highlight how the delay might be detected in such experiments. We also show that the existence of the delay could explain an observed discrepancy in the measurement of mutation rates, and demonstrate that one of our models provides a superior fit to experimental data. Our work exposes how molecular details at the intracellular level can have a direct effect on evolution at the population level.

## Introduction

The emergence of resistance to drugs is a significant problem in the treatment of diseases such as cancer [1], and viral [2] and bacterial infections [3]. In infections with high pathogen load, the occurrence of *de novo* genetic mutations leading to resistance is a significant problem [4]; examples include endocarditis infections caused by *Staphylococcus aureus* [5, 6], *Pseudomonas aeruginosa* infections of cystic fibrosis patients [7, 8], as well as *Burkholderia dolosa* [4, 9] infections.

The emergence and spread of resistant variants in populations of pathogenic cells has received much experimental [10–14] and theoretical attention [15–18]. However, most mathematical models assume that a genetic mutation immediately transforms a sensitive cell into a resistant cell [19–24]. In reality, a new allele (genetic variant) must be expressed to a sufficient level before the cell becomes phenotypically resistant. The time between the occurrence of a genetic mutation and its phenotypic expression is called *phenotypic delay*. This is also referred to as delayed phenotypic expression, phenotypic lag, cytoplasmic lag or phenomic lag.

Phenotypic delay was first observed in 1934 by Sonnenborn and Lynch when studying the effect of conjugation on the fission rate of *Paramecium aurelia* [25]. Phenotypic delay was further studied during the 1940s and 1950s, both theoretically [26] and experimentally [27, 28]. Interestingly, in their hallmark work on the randomness of mutations in bacteria [29], Luria and Delbrück discussed the possible effect of a phenotypic delay on the estimation of mutation rates. However, interest in phenotypic delay waned for the next seventy years, mostly because experimental data failed to reveal evidence for such delay [29, 30]. However, Sun et al. [31] recently demonstrated the existence of a phenotypic delay of 3-4 generations in the evolution of resistance of *Escherichia coli* to the antibiotics rifampicin, nalidixic acid and streptomycin. Sun et al. attributed this delay to effective polyploidy.

Here, we generalize these observations and also investigate other mechanisms that may lead to phenotypic delay. We consider three mechanisms: (i) effective polyploidy as in Sun et al. [31], (ii) the dilution of sensitive molecules targeted by the drug, and (iii) the accumulation of resistance-enhancing molecules. We speculate on the relevance of these mechanisms for different antibiotics in Table 1.

*Effective polyploidy* refers to the fact that a single cell can contain multiple copies of a given gene. This can be due to gene duplication events or carriage of multicopy plasmids; it also occurs in fast-growing bacteria, which initiate new rounds of DNA replication before the previous round has finished, allowing for a shorter generation time than the time needed to replicate the chromosome [32–34]. Since a *de novo* resistance mutation happens in only one of the multiple gene copies, it may take several generations before a cell emerges in which all gene copies contain the mutated allele. Until then, sensitive and resistant variants of the target protein coexist in the cell. A phenotypic delay occurs when the resistance mutation is recessive, i.e., the sensitive variant must be replaced by the resistant variant for the cell to become

**Table 1. Postulated mechanism of phenotypic lag for different antibiotics discussed in this work.**

| Antibiotic | Target | Mechanism of resistance | Postulated phenotypic lag model |
|---|---|---|---|
| rifampicin | RNA polymerase | target mutation (*rpoB*), recessive | dilution+polyploidy |
| fluoroquinolones | DNA topoisomerases | target mutation (*gyrA*, *gyrB*, *parC*), recessive | dilution+polyploidy |
| polymixins | lipo-poly-saccharide (LPS) | mutations in enzymes modifying the structure of LPS | dilution+polyploidy |
| beta-lactams | enzymes in cell wall synthesis | inactivation by beta-lactamase, dominant | accumulation |
| tetracycline | ribosomes | efflux upregulation, production of a protective protein | accumulation |
| many antibiotics | different targets | upregulation of efflux pumps, dominant | accumulation |

resistant. This is the case for antibiotics which form toxic adducts with their targets [35, 36]. Examples are quinolones that lock the enzyme DNA gyrase onto the DNA and prevent DNA replication [37], and polymixins that bind to lipids in the outer membrane which causes membrane perforation [38, 39]. Effective polyploidy also changes the per-cell mutation rate, because it alters the number of gene copies per cell [31]. However, as shown both by Sun et al [31] and in this paper, it does not alter the distribution of mutant numbers that are observed in fluctuation tests.

The *dilution mechanism* also assumes the mutation to be recessive, but in contrast to the polyploidy mechanism it focuses on the removal of the sensitive target protein through the process of cell growth and division. As a mutated cell grows, the resistant version of the protein accumulates; a subsequent division creates two cells in which the fraction of the sensitive variant is less than in the parent cell. Even if the relevant gene is present only in a single copy (ruling out effective polyploidy), there may still be a considerable delay if the number of sensitive proteins to dilute out is large before resistance can be established.

The *accumulation mechanism* posits that sufficient copies of the resistant variants of a protein must be produced to cause resistance. This is likely to apply to mutations that enhance the expression of drug efflux pumps [40], $\beta$-lactamase enzymes that hydrolyse $\beta$-lactam antibiotics [41, 42], or mutations that protect ribosomes from tetracycline [43], hence restoring the active ribosome pool [44]. In these cases, a phenotypic delay could emerge due to the time required for the resistance-enhancing protein to accumulate in the cell to a level high enough to cause resistance.

We first analyse the three mechanisms using computer simulations and analytic calculations. We find that the accumulation of resistance-enhancing molecules only leads to phenotypic delay within a limited parameter range, while effective polyploidy and the dilution of sensitive molecules lead to phenotypic delay for a broad range of parameters. We also show that while the effective polyploidy mechanism does not affect the probability that a population survives antibiotic challenge, dilution of sensitive protein leads to decreased probability of survival under drug treatment.

We then investigate the possibility of detecting a phenotypic delay experimentally. We first show that the dilution mechanism would cause an underestimation of mutation rates in Luria-Delbrück fluctuation tests compared to the true genetic rate of mutations. In a fluctuation test, one measures the distribution of mutant numbers in replicate populations that have been allowed to grow and evolve for a fixed number of generations. The mutation rate is then estimated by fitting a population dynamics model to the experimental distribution [29, 45]. Our result is consistent with the fact that the mutation rate of *Escherichia coli* obtained in fluctuation tests has been found to be an order of magnitude smaller than the rate obtained by DNA sequencing [46].

We also show that the dilution mechanism subtly alters the shape of the Luria-Delbrück distribution of mutant numbers. Discrepancies between the shapes of the experimental and theoretically predicted mutant number distributions have been observed since the original experiments of Luria and Delbrück [26, 29, 30, 47], but have never been satisfactorily explained. Using an experimental data set reported by Boe et al. [47] for *E. coli* and for fluoroquinolone antibiotics, we show that a mathematical model that includes the dilution mechanism fits the data better than the no-delay Luria-Delbrück model, thus providing indirect evidence for the existence of this type of phenotypic delay in the *de novo* evolution of resistance to fluoroquinolones.

## Results

### Modeling the emergence of phenotypic delay

To explore the characteristic features of the three different phenotypic delay mechanisms—dilution of sensitive molecules, effective polyploidy, and accumulation of the resistant variant—we first simulate an idealised mutagenesis experiment (Fig 1a). We suppose that at the start of the experiment a population of sensitive bacteria is exposed to a mutagen (e.g., UV radiation [48, 49]) which instantaneously induces mutations in a small fraction of the cells [50, 51]. Cells immediately begin to express the mutated allele, but because of the existence of phenotypic delay, they remain sensitive to the antibiotic for some time; phenotypically resistant cells emerge only after a few generations.

We investigate the emergence of resistance in two different ways. The first approach is to follow a random lineage, starting from a single mutant bacterium (i.e., at each division we follow one of the randomly selected daughter cells) and to measure the waiting time before a phenotypically resistant cell emerges *in that lineage* (S1 Fig). The second approach is to track the entire population post mutagenesis, and examine the waiting time before the first phenotypically resistant cell emerges *in the whole population*. Fig 1b shows the conceptual difference between these two approaches.

**Dilution of antibiotic-sensitive molecules.** Even in the absence of polyploidy, if the resistance mutation is recessive such that a small number of sensitive target molecules are enough to cause antibiotic sensitivity, a phenotypic delay can arise from the time taken to replace sensitive target molecules by resistant ones. To model this, we assume that each cell has a number $n$ of target molecules that are initially sensitive. Once a mutation has happened, production of sensitive molecules ceases and only resistant molecules are produced. We suppose that, upon cell division, the $n$ molecules are partitioned stochastically without bias between the two daughter cells (Fig 1c). For simplicity, in this work cells are considered phenotypically resistant only when they contain no sensitive molecules, i.e., the number of sensitive molecules that need to be diluted out is $n$.

For this mechanism, the length of the phenotypic delay increases approximately logarithmically with the number $n$ of sensitive molecules that need to be diluted for resistance to emerge (Fig 1d). To understand this, suppose momentarily that $n$ is a power of 2 and stochasticity can be neglected so that each daughter cell receives exactly half the number of molecules of the parent cell. Then for any lineage stemming from a genetically mutant cell, the number of inherited sensitive molecules will be $2^{n-1}$, $2^{n-2}$, ..., as the generations progress. After $\log_2 n$ generations all cells will have a single sensitive molecule and hence the first phenotypically resistant cell will then emerge after $1 + \log_2 n$ generations. In this deterministic setting, $1 + \log_2 n$ will also be the number of generations for the population to become resistant.

In the more realistic case of stochastic segregation of molecules, the probability of resistance along a random lineage after $g$ generations is approximately $\exp(-2^{-g}n)$ (S1 Text, Section 1.1).

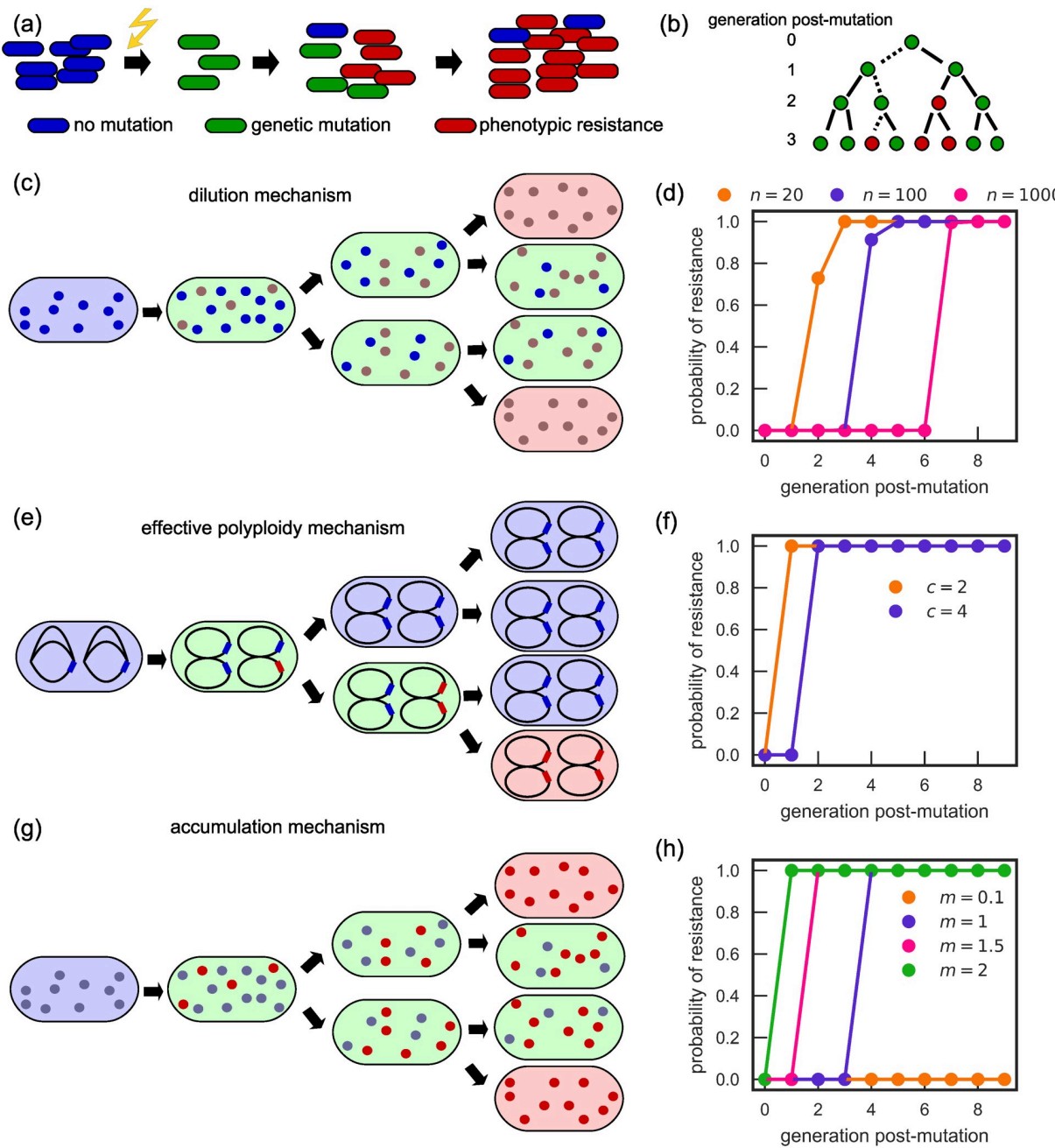

**Fig 1. Models of phenotypic delay.** (a) Schematic representation of a simulated experiment, in which a mutagen (e.g., UV radiation) induces resistant mutations at a particular moment in time. Mutants initially remain sensitive to the antibiotic, only becoming resistant after a few generations. (b) Two ways of determining the time to resistance: tracking a single random lineage (dotted line), and tracking the whole population. In this example, resistance emerges in generations 3 and 2, respectively. (c) The dilution mechanism: blue/brown dots denote sensitive/resistant variants of the target molecule. When a wild-type cell (blue) mutates, it initially remains sensitive (green) and becomes resistant (red) when all sensitive molecules are diluted out. (d) Probability that at least one cell in an exponentially growing population starting with 100 newly genetically mutated cells is phenotypically resistant (dilution model) as a function of the number of generations since the genetic mutation (dots: simulation; lines: theory). Phenotypic delay increases with the number of molecules $n$ to be diluted. (e) The effective polyploidy mechanism: chromosomes are represented as black ellipses, with a sensitive/resistant allele marked blue/red. (f) Same as in (d) but for the effective polyploidy mechanism. Phenotypic delay increases with ploidy $c$. (g) The accumulation mechanism: blue/red dots denote sensitive/resistant mutants of the resistant-enhancing molecule. Cells become resistant (red) when the cell contains enough resistant molecules. (h) Same as in (d) but for the accumulation model. Phenotypic delay decreases with increasing ratio $m$ of the number of molecules produced during cell cycle and the number of molecules required for resistance.

Hence the probability of resistance emerging in a lineage is negligible until generation $g$ set by $2^g \approx n$, when the probability rapidly rises to 1. Therefore, in line with our deterministic reasoning, resistance along a random lineage will emerge after $g \approx \log_2 n$ generations. Interestingly, however, we obtain a different result for the probability that the population as a whole produces at least one resistant cell. If we start from $x$ genetically mutated cells in the population, the first phenotypically resistant cell in the population emerges, on average, after an approximate time $1 + \log_2(n/\log(xn))$ (S1 Text, Section 1.1). We can also calculate the resistance probability through a recursion relation (S1 Text, Section 1.1); the results fully reproduce the simulations (Fig 1d). The emergence of resistance at the population level is thus accelerated compared to what one would obtain based on deterministic dilution. We have assumed for simplicity that each of the $x$ cells initially has the same number $n$ of sensitive molecules; this is only a crude approximation for real bacteria. An extended model in which molecules are distributed in a biased way between the two daughter cells, inspired by recent evidence on accumulation of membrane proteins in the daughter cell with the older pole [52–55], leads to a similar result (S1 Text, Section 3). However, the bias decreases slightly the phenotypic delay at a population level (S3 Fig); this is because the bias creates lineages which are low in the number of resistant molecules.

**Effective polyploidy.**   Rapidly dividing bacteria can become effectively polyploid when they initiate a round of DNA replication before the previous round has finished; this leads to the presence of multiple copies of at least some parts of the chromosome [32] (Fig 1e). Crucially, the degree of polyploidy (number of gene copies) depends on the bacterial growth rate, as well as on other factors such as the genetic locus. To model phenotypic delay caused by effective polyploidy, we assume that each cell has a number $c$ of chromosome copies that is growth-rate dependent according to the well-established Cooper-Helmstetter model of *E. coli* chromosome replication [32] (Methods). Each chromosome copy contains a single allele, encoding the antibiotic target, that can be either sensitive or resistant. Initially all chromosomes have the sensitive allele but a mutation changes one allele from sensitive to resistant. We then simulate the process of DNA replication and cell division, taking account of the fact that duplicated resistant alleles are co-inherited—for example, if a cell has two chromosome copies, one with a resistant allele and the other with a sensitive allele, then upon replication and division, one daughter cell will have two sensitive alleles and the other daughter cell will have two resistant alleles [33] (Methods). We assume that a cell becomes phenotypically resistant when none of its chromosomes contain the sensitive allele (i.e., the resistant allele is assumed to be recessive). In this model, the waiting time until a cell acquires a full suite of resistant chromosomes is $\log_2 c$ generations (Fig 1f). This phenotypic delay time is the same whether we track a given lineage or the entire population (since it is deterministic). However, resistance will not occur in all lineages; of the $c$ lineages descended from the original mutant cell, resistance will eventually occur in only one of them [31] (S1 Fig).

We note that effective polyploidy generally causes a shorter delay than dilution of sensitive molecules: 2 to 3 generations for rapidly growing bacteria ($c = 4$ or $8$ [31, 32]), versus 5 generations for the dilution mechanism (assuming $n \approx 500$, which is typical for the gyrase enzyme targeted by fluoroquinolones [56, 57]). The transition in the probability of resistance as a function of time is also sharper for effective polyploidy than for the dilution mechanism in which stochasticity of the segregation process smooths out the transition (compare Fig 1d and 1f). Finally, for effective polyploidy, we expect only one in every $c$ lineages to become resistant, while for dilution of sensitive molecules, all lineages will eventually become resistant.

**Accumulation of resistance-enhancing molecules.**   Phenotypic delay can also emerge due to the time needed to accumulate resistance-enhancing molecules to a sufficiently high level (Fig 1g). To model this mechanism, we suppose that during each cell cycle a genetically

resistant cell produces $M_p$ resistance-enhancing molecules, which are randomly distributed between daughter cells at division. A cell becomes resistant when it has $M_r$ or more resistance-enhancing molecules. Interestingly, considering either a single lineage (S1 Fig) or the entire population (Fig 1h), we find that phenotypic delay emerges only within a limited parameter range: $1 \lesssim m \lesssim 2$, where $m = \frac{M_p}{M_r}$ is the ratio of the number of molecules produced during a cell cycle and the number of molecules needed for resistance. The origin of this limited parameter range is most easily explained by considering a single lineage. Tracking a lineage arising from a single mutant cell, the cell in the $g$th generation will be born with an average of $M_p(1 - 2^{-g})$ molecules (S1 Text, Section 1.2). The steady-state number of molecules (found by taking $g \rightarrow \infty$) is $M_p$. Thus if $m < 1$, the steady state number of molecules will be always smaller than the minimum required number $M_r$, and the lineage will never become phenotypically resistant. Conversely, if $m > 1$, phenotypic resistance will emerge after approximately $\tau = -\log_2(1 - 1/m)$ generations when the average number of resistance-enhancing molecules exceeds $M_r$. But for the delay to be at least one generation long ($\tau \geq 1$), we require $m \leq 2$. Considering now the scenario where we track the entire population, we again expect the steady-state molecule number $M_p$ to be rapidly reached by all cells, so that there will be no phenotypic resistance for $m < 1$. Further, if resistance does emerge (for $m > 1$), it will do so more quickly in the entire population than along the random lineage (as resistance may be acquired in any lineage). We thus expect an even tighter upper bound on the value of $m$ for phenotypic delay to manifest itself on the population level in this model.

Since our analysis shows that, for this mechanism, phenotypic delay only emerges in a narrow parameter range, we conclude that the accumulation of resistance-enhancing molecules is unlikely to be biologically relevant in causing phenotypic delay. Therefore we do not explore this mechanism further.

## Combining effective polyploidy and dilution

In reality, for a recessive resistance mutation, we expect both the effective polyploidy and dilution mechanisms to contribute to the phenotypic delay. To understand the implications of this, we simulated a model combining the two mechanisms, tracking the emergence of resistance at a single-cell and population level. Our simulations predict a phenotypic delay with characteristics of both mechanisms (Fig 2).

Focusing first on a single lineage (Fig 2a and 2b), we observe that the long-term probability of phenotypic resistance depends on the ploidy $c$, tending to $1/c$, as expected for the effective polyploidy mechanism, while the approach to this value is gradual as expected for the dilution mechanism. Combining both mechanisms increases the length of the delay compared to either mechanism acting in isolation.

Following Sun et al. [31], we also calculate the phenotypic penetrance, defined as the proportion of genetic mutants which are phenotypically resistant in the entire population. The expected phenotypic penetrance (see S1 Text Section 1.3 for derivation) is:

$$\begin{cases} 0 & 0 \leq g < \log_2 c, \\ (1 - 2^{-g})^n \prod_{i=0}^{\log_2 c - 1} \left(1 - 2^{-(g-i)}\right)^{n(1-2^i/c)} & \log_2 c \leq g. \end{cases} \quad (1)$$

Note that $n = 0$ corresponds to only the effective polyploidy mechanism, while $c = 1$ corresponds to only the dilution mechanism being present. The piecewise form of Eq (1) arises because no cell can become phenotypically resistant until all its chromosomes have the resistant allele. Fig 2d shows that the phenotypic penetrance predicted by Eq (1) increases gradually

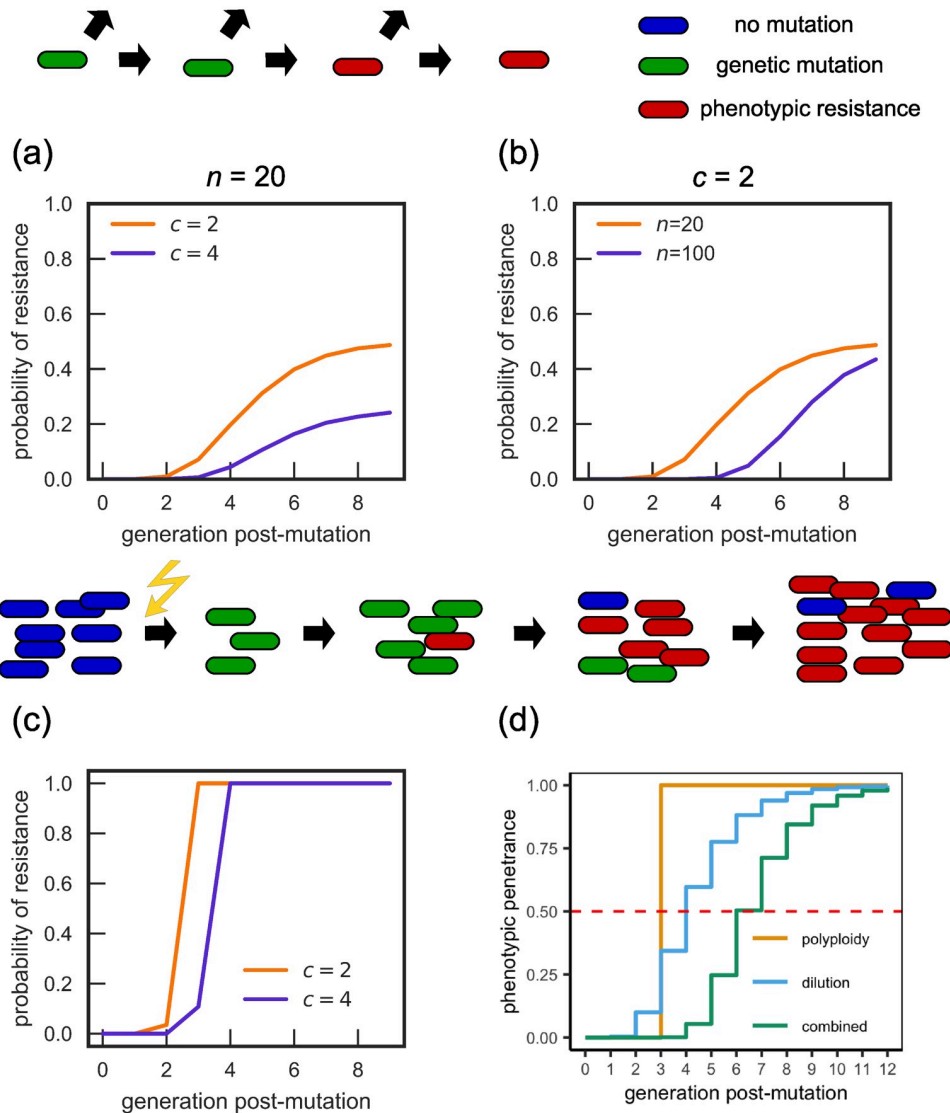

**Fig 2. Combined effect of the dilution and effective polyploidy mechanisms.** (a-b) Probability of resistance of a single mutated cell. While the long-term probability is defined by the effective polyploidy, short-term behaviour is determined by the dilution mechanism, leading to longer phenotypic delays than the effective polyploidy mechanism would produce. (c) Population-level probability of resistance versus the number of generations from the mutation event, for $n = 20$. The combined mechanism leads to smoother curves than the effective polyploidy mechanism and longer delays than for either mechanism individually. (d) Phenotypic penetrance (ratio of phenotypically resistant to genetically resistant cells, obtained from Eq (1)) for the different mechanisms, for $c = 8$, $n = 8$. The dashed red line indicates when the phenotypic penetrance surpasses 1/2, which is the threshold used by Sun et al. [31] to define the emergence of phenotypic resistance. With this definition, the dilution mechanism plus effective polyploidy doubles the delay (generation 6 as opposed to generation 3 compared to effective polyploidy alone).

with time (characteristic of the dilution mechanism) but with a delay determined by effective polyploidy.

We now return to computer simulations to study the emergence of resistance on the population level following mutagenesis (the thought experiment from Fig 1a), for the combined delay mechanisms. In general, both the ploidy $c$ and the number of antibiotic target molecules per cell $n$ will depend on the doubling time $t_d$ (or growth rate) of cells. To be more

specific, we consider resistance of *E. coli* to fluoroquinolone antibiotics, that arises through mutations in DNA gyrase (protein targeted by the antibiotic). Gyrase abundance as a fraction of the proteome (i.e. gyrase concentration in the cell) has been found to be independent of the growth rate [58]. We therefore assume that the number $n$ of gyrases per cell is proportional to the cell volume $V$. We model the volume as $V \propto 2^{\lambda/\lambda_0}$, where $\lambda = (\ln 2)/t_d$ is the growth rate and $\lambda_0 = 1\text{h}^{-1}$ [59–62], and we model polyploidy using the Cooper-Helmstetter model [32] (see Methods and model for details). Suppose that for slow-growing cells ($t_d =$ 60 min), $c = 2$ and $n = 20$. Then, for fast-growing cells ($t_d = 30$ min), we have $c = 4$ and $n = 40$. Note that here we do not assume realistic values of $n$ because the minimum number $n_r$ of poisoned sensitive gyrase molecules required to inhibit growth is probably much higher than $n_r = 1$ assumed in the model. $n$ should be therefore interpreted more correctly as the number of "units" of gyrase, with one unit equivalent to $n_r$ molecules. Fig 2c shows that the phenotypic delay is longer for the fast-growing population, and that this is mostly caused by the increase in the number of molecules $n$ (S4 Fig). We also observe that protein dilution leads to a smoother transition between sensitivity and resistance than the transition due to effective polyploidy alone.

## The dilution mechanism, but not effective polyploidy, affects the probability of clearing an infection

To understand better the practical significance of phenotypic delay, we simulated antibiotic treatment of an idealised bacterial infection (Fig 3). We assume for simplicity that, before treatment, the population of bacteria grows exponentially in discrete generations, and cells mutate with probability $\mu = 10^{-7}$ per cell per replication. When the population size reaches $10^7$, an antibiotic is introduced; this causes all phenotypically sensitive bacteria to die, leaving only the phenotypically resistant cells (Fig 3b). We are interested in the probability that the bacterial infection survives the antibiotic treatment, a concept closely related to evolutionary rescue probability, i.e., the probability that cells can survive a sudden environmental change thanks to an adaptive mutation [31, 63, 64]. Since sensitive cells do not reproduce in our simulations in the presence of the antibiotic, survival can only be due to pre-existing mutations (standing genetic variation).

We first consider the effective polyploidy model, with ploidy $c$ controlled by the doubling time $t_d$. In agreement with Sun et al. [31], we find that $t_d$ has no effect on the survival probability (Fig 3c). This is due to a cancellation of two effects: the increased number of gene copies increases the per-cell chance of genetic mutation, but also increases the length of the phenotypic delay (see Section 2.2.1 of the SI of Ref. [31] for a mathematical derivation). In contrast, phenotypic delay caused by the dilution of sensitive molecules does affect the survival probability (Fig 3d). The survival probability strongly depends on $n$, and decreases significantly from 0.69 for $n = 0$ to 0.06 for $n = 100$.

We also simulated the mixed case where both the effective polyploidy and dilution mechanisms are combined, with ploidy $c$ and molecule number $n$ determined by the doubling time $t_d$ as described in Sec. *Combining effective polyploidy and dilution*. In this case the survival probability does depend on the doubling time (Fig 3e; blue line). This is mostly caused by the change in the molecular number $n$ as a function of doubling time. If we neglect the dependence of $n$ on $t_d$, the effect is much smaller, although there is still some dependence on $t_d$ because the rate of resistant protein production depends on the resistant gene copy number, which increases en route to the full suite of resistant chromosomes (S4 Fig).

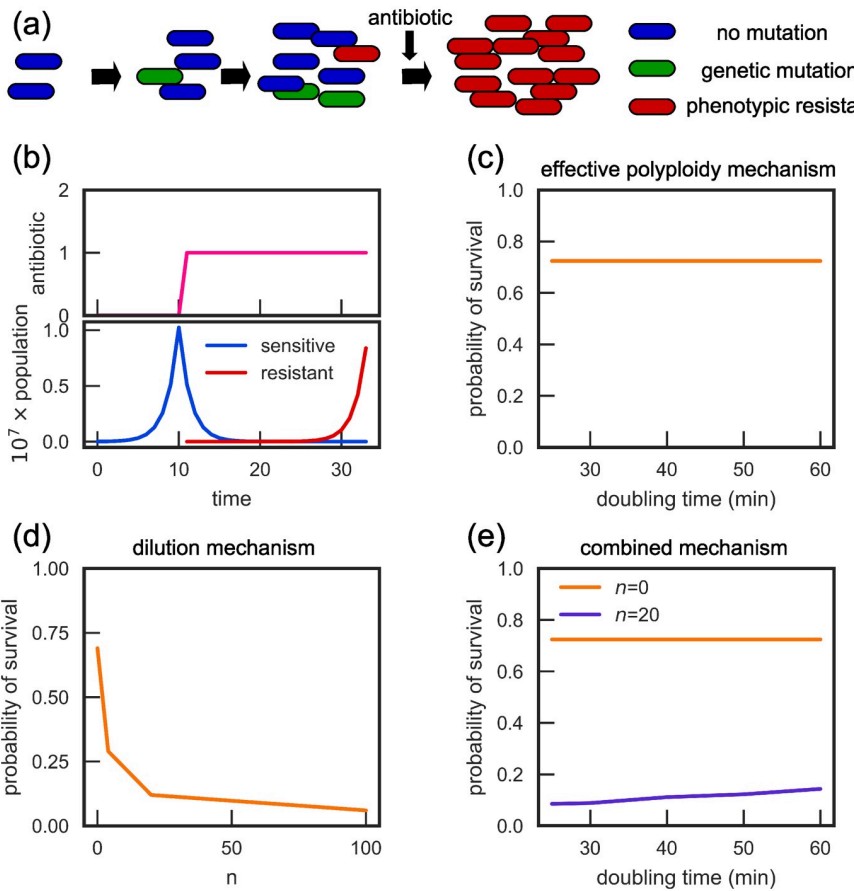

**Fig 3. Phenotypic delay decreases the probability of a bacterial infection surviving antibiotic treatment.** (a-b) A schematic of the simulated infection: a population of exponentially replicating sensitive cells is exposed to an antibiotic when the population reaches $10^7$ cells. Only phenotypically resistant cells survive the antibiotic. Time and antibiotic concentration in panel (b) have arbitrary units. (c) The probability of survival for the effective polyploidy mechanism is independent of the doubling time (and hence the ploidy). (d) For the dilution mechanism, the probability of survival decreases with the number of molecules $n$ which need to be diluted out before the cell becomes phenotypically resistant. (e) In a combined dilution-and-effective polyploidy model, the survival probability increases with the doubling time.

## Phenotypic delay due to dilution changes the Luria-Delbrück distribution and biases mutation rate estimates

The scenario discussed in the previous section is equivalent to the Luria-Delbrück fluctuation test [29, 65], which has been extensively studied theoretically [45, 66–73]. In the fluctuation test, a small number of sensitive bacteria are allowed to grow until the population reaches a certain size. The cells are then plated on a selective medium (often an antibiotic) to reveal the number of mutated bacteria in the population. The distribution of the number of mutants (measured over replicate experiments) is termed the Luria-Delbrück distribution. This distribution has a power-law tail caused by mutational "jackpot" events [29, 65, 72] in which rare, early-occurring mutants produce many descendants in the population. The fluctuation test, fitted to corresponding mathematical models, is widely used to estimate mutation rates in bacteria. Here, we discuss the effect of phenotypic delay on the Luria-Delbrück distribution and on the resulting mutation rate estimate.

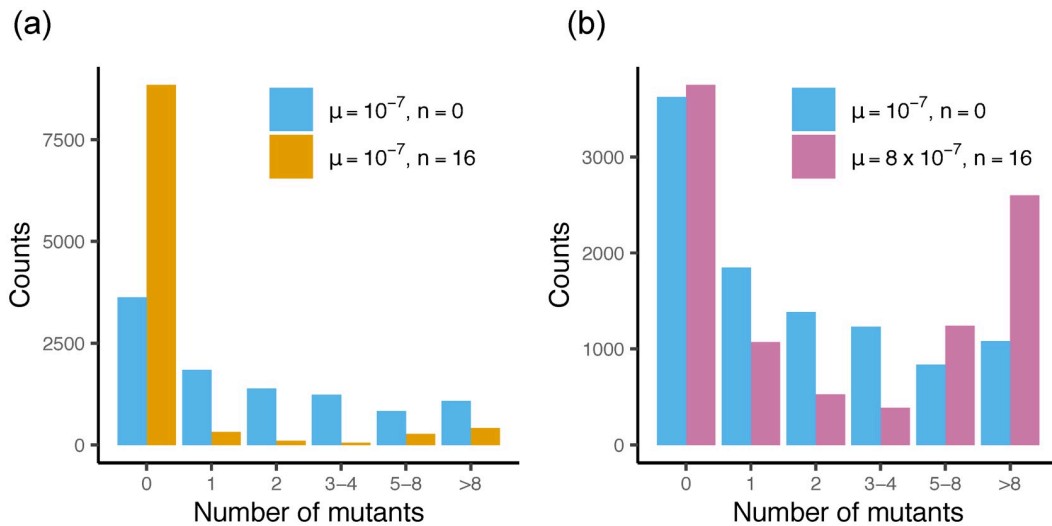

**Fig 4. The dilution model affects the probability distribution of the number of resistant cells.** The frequency of mutants for a simulated fluctuation test with 10,000 samples, for the model with $n = 0$ (no delay) and $n = 16$. (a) Distributions for both models for a fixed $\mu = 10^{-7}$. (b) Distributions for the case when $\mu$ in the dilution model has been adjusted to minimize the difference to the no-delay model (values in the inset).

First, we note that phenotypic delay caused by effective polyploidy alone does not affect the Luria-Delbrück distribution. As discussed in the previous section, this is due to an exact cancellation of two effects: increased ploidy leads to more mutations per bacterium but also a longer phenotypic delay. In contrast, the dilution model does alter the Luria-Delbrück distribution. Fig 4a shows that, for a fixed mutation probability $\mu$ and fixed initial and final population sizes, phenotypic delay due to dilution causes an increase in the number of replicate experiments yielding zero resistant mutants, and a decrease in the number of experiments yielding intermediate numbers of resistant mutants. The number of experiments yielding very large numbers of mutants, due to jackpot events, is less affected by the delay—this is because mutants that arise early will have had sufficient time to dilute out the sensitive molecules and become phenotypically resistant before being exposed to the antibiotic. Hence the dilution model also leads to a similar scaling (proportion of replicates yielding at least $x$ mutants is $\propto x^{-1}$ for large $x$) as for the Luria-Delbrück distribution (S8 Fig).

From a practical point of view, the mutation probability is often unknown and the fluctuation test is used to estimate it. To investigate the effect of phenotypic delay on the estimated mutation probability, we simulated the fluctuation test for the dilution model with $n = 16$, for a range of mutation probabilities. We compared the resulting mutant number distributions to that obtained in an equivalent simulation without phenotypic delay, with mutation probability $\mu = 10^{-7}$. Using a genetic algorithm [74] to minimize the $L_2$ norm between the distributions with and without phenotypic delay, we found that the phenotypic delay model required a much larger mutation probability ($\mu = 8 \times 10^{-7}$) to reproduce the distribution of the no-delay model. This suggests that neglecting phenotypic delay when fitting theory to fluctuation test data could significantly underestimate the true mutation probability. We also note that the "closest match" distributions with and without phenotypic delay are not exactly identical (Fig 4b). The model with phenotypic delay leads to a larger number of jackpot events (as might be expected since the mutation probability is higher) and a reduced number of replicates with few mutants, consistent with suppression of late-occuring mutants by the phenotypic delay.

Our result could explain an apparent discrepancy between mutation probabilities estimated by different methods. In particular, Lee et al. measured the mutation probability of *E. coli* using both fluctuation tests (with the fluoroquinolone nalidixic acid as selective agent) and whole-genome sequencing [46]. The fluctuation test underestimated the mutation probability by a factor of 9.5; Lee et al. suggested that this could be caused by phenotypic delay [46]. To see whether our dilution model could explain this, we simulated the 40-replicate, 20 generation fluctuation test experiment of Lee et al. [46], using the mutation probability as estimated by whole-genome sequencing ($\mu = 3.98 \times 10^{-9}$, total for all mutations producing sufficient resistance to nalidixic acid), for differing values of the number $n$ of target "units" ("effective" gyrase molecules). For each $n$ we simulated 1000 realisations of the 40-replicate experiment, and for each realisation we estimated the mutation probability under the no-delay model using the maximum likelihood method [45] (the same as used by Lee et al.) implemented in the package flan [75]. This procedure correctly reproduced the mutation probability for data from simulations without delay ($n = 0$; S6 Fig). For the model with delay, the maximum likelihood fit returned a mutation probability that was lower than the true one (Fig 5a); the discrepancy increased with the phenotypic delay. To obtain an apparent mutation probability that is underestimated by a factor of 9.5, as observed by Lee et al. [46], we require $n \approx 30$; i.e. roughly 30 sensitive 'units' of the antibiotic target must be diluted out before a cell becomes phenotypically resistant. Thus, while our simulations do not prove that phenotypic delay is responsible for the discrepancy observed by Lee et al., they suggest that it is a plausible explanation.

## Mutant number distributions may support the existence of phenotypic delay

Our results suggest that a phenotypic delay caused by dilution produces a characteristic (though small) change in the shape of the observed mutant number distribution (Fig 4b). This deviation should, in principle be detectable in experiments. To check this, we used the dataset of Boe et al. [47] who performed a 1104-replicate fluctuation test, using the bacterium *E. coli* with the fluoroquinolone antibiotic nalidixic acid as the selective agent. Nalidixic acid targets DNA gyrase. As explained in Sec. *Combining effective polyploidy and dilution*, we expect that a small number of wild-type DNA gyrases should be enough for a bacterial cell to be sensitive to the antibiotic, suggesting that phenotypic delay via gyrase dilution may be likely. Boe et al. [47] report an unsatisfactory fit of their mutant number distribution data to the theoretical predictions of two different variants of the Luria-Delbrück model (the Lea-Coulson and Haldane models); in comparison to these models, Boe et al. observed too many experiments yielding either no mutants or a high number of mutants (greater than 16), and a dearth of experiments resulting in intermediate mutant counts (1-16). Qualitatively, this seems to be consistent with our expectations for the dilution model (Fig 4).

To see if the dilution model of phenotypic delay indeed provides a superior fit to Boe et al's data, we used an approximate Bayesian computation (ABC) approach [76] (Methods). We simulated a 1104-replicate fluctuation experiment $10^4$ times, for the models with and without delay, with initial and final population sizes of $1.2 \times 10^4$ and $1.2 \times 10^9$ matching those of Boe et al. [47]. We then determined the posterior Bayesian probability that the experimental data is generated by the delay model as opposed to the no-delay model, and tested the validity of our approach using synthetic data (Methods and models). We find that the probability of the experimental data coming from the model with phenotypic delay is 0.97, as opposed to the model without phenotypic delay (Fig 5b). We thus conclude that the Boe et al. data supports the existence of phenotypic delay caused by the dilution mechanism.

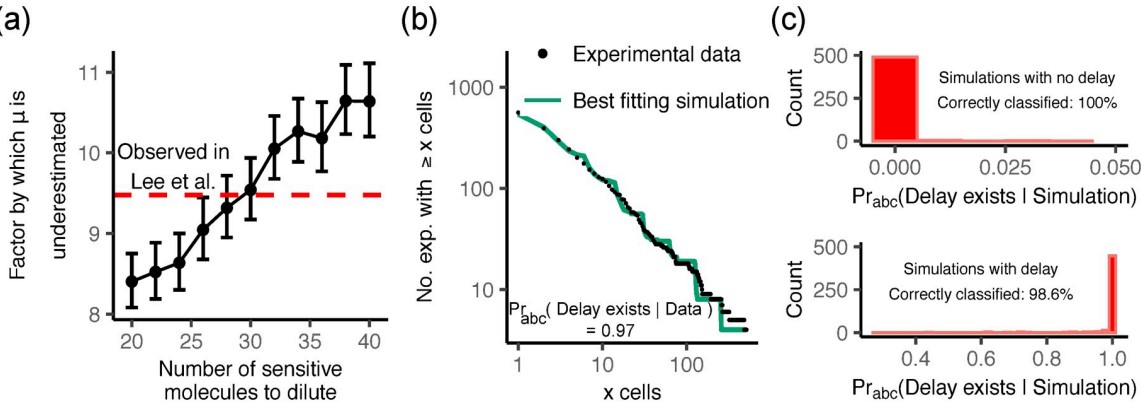

**Fig 5. Phenotypic delay due to the dilution mechanism explains observed discrepancy in mutation rates and provides superior fit to fluctuation experiment data.** (a) We simulated the fluctuation experiment of Ref. [46], where the authors report a factor of 9.5 difference between the values of $\mu$ obtained by DNA sequencing and fluctuation tests. For each $n$ we simulated 1000 experiments with the sequencing-derived mutation probability $\mu = 3.98 \times 10^{-9}$ and then used the same estimation procedure as Ref. [46] to infer $\mu$ assuming no delay exists. $n = 30$ sensitive molecules are required to account for the discrepancy observed. Error bars are $1.96 \times$ standard error. (b) The experimental cumulative mutant frequency distribution reported by Boe et al. [47] (black points) and the best-fit simulated distribution (green line) for the dilution phenotypic delay model. The staircase-like shape of the simulated distribution is caused by the fixed division time and strictly synchronous division of the mutated cells. (c) Histograms of the probability of the delay model obtained by applying the approximate Bayesian computation scheme to simulated data. Our classification algorithm correctly discriminates between the models.

## Discussion

Quantitative models for *de novo* evolution of drug resistance are an important tool in tackling bacterial antimicrobial resistance, as well as viral infections and cancer. However, our quantitative understanding of how resistance emerges is still limited. The possibility of a phenotypic delay between the occurrence of a genetic mutation and its phenotypic expression has long been discussed [25–29], but its relevance for bacterial evolution has been questioned until recently [31]. Here, we have used computer simulations and theory to study the effects of phenotypic delay on the emergence of bacterial resistance to antibiotics. We investigated three different mechanisms that could lead to phenotypic delay: (i) dilution of antibiotic-sensitive molecules, (ii) effective polyploidy, and (iii) accumulation of resistance-enhancing molecules. We observe that the third mechanism only leads to phenotypic delay under a limited range of parameters, which makes it unlikely to be biologically relevant. The other two mechanisms have different "control parameters" (the degree of ploidy $c$ versus the number of target molecules $n$) and different effects on the population dynamics. In particular, we show that protein dilution, but not effective polyploidy, can affect the probability that a growing population survives antibiotic treatment. This in turn can bias the estimated mutation rate in a Luria-Delbrück fluctuation test. Effective polyploidy does not play a role here because of two cancelling effects: increased ploidy increases the number of mutations per cell in the growing population, but also increases the length of the phenotypic delay. These effects counterbalance such that the Luria-Delbrück distribution remains unaffected [31].

### Effect of the dilution mechanism on the lineage/population survival probability

We have shown that the various mechanisms affect the survival of whole populations, and of random lineages, in different ways. In the case of effective polyploidy, the duration of the phenotypic lag is the same for a random lineage as it is for the entire population. However, only

one in $c$ lineages becomes resistant and can survive antibiotic treatment. In contrast, in the dilution mechanism every lineage becomes resistant and survives, as long as the time before antibiotic exposure is much longer than the phenotypic lag. However, the length of the phenotypic lag for each lineage is now a random variable. The time to resistance at the population level is thus determined by the shortest phenotypic lag among all the lineages.

## Effect of the dilution mechanism on fluctuation test data

Luria-Delbrück fluctuation tests remain the standard microbiological method for estimating mutation rates, yet it has often been noted that the measured distributions of mutant numbers are not precisely fit by the theoretical distribution [26, 29, 30, 47]. A comparison with a more direct approach (DNA sequencing) suggests that fluctuation tests can significantly underestimate mutation rates [46]. Although phenotypic delay has been suggested as a possible explanation for these effects [29, 46], our study is the first to investigate in detail how specific mechanisms of phenotypic delay alter the shape of the Luria-Delbrück distribution, and to demonstrate that it can indeed produce a mutation rate estimate that is biased in the same way as that observed experimentally [46]. We also show that the simulated distribution of mutant numbers from the dilution model fits the experimental fluctuation test data of Boe et al. [47] better than the standard model without phenotypic delay. We note that this result should however be taken cautiously. Boe et al.'s experimental protocol is not ideal for detecting phenotypic delay: for example, their bacterial cultures were allowed to reach stationary phase before plating. Moreover, our work shows that while phenotypic delay due to dilution affects the mutant number distribution, the change is subtle, requiring many replicate experiments to produce statistically significant results. While the usual number of replicates in a fluctuation test is less than 100, recent developments in automated culture methods should make it possible to run fluctuation tests with many more replicates, which may provide a way to probe the effects of phenotypic delay on the Luria-Delbrück distribution in more detail.

## From molecular detail to evolutionary population dynamics

Our work presents an example of how molecular details at the intracellular level (here, protein dilution and the details of DNA replication) can have a direct effect on evolution at the population level [77–79]. This observation complements other work showing, for example, that molecular processes such as transcription and translation affect population-level distributions of protein numbers [80, 81] and that noise in gene expression can directly affect the survival of populations in a fluctuating environment [82].

Importantly, both the effective polyploidy mechanism and the dilution mechanism cause a phenotypic delay only if the resistance mutation is *recessive*. For effective polyploidy this means that a cell must contain only resistant alleles in order to be phenotypically resistant, while for the dilution mechanism we have assumed that sensitive target molecules need to be diluted out (or otherwise removed). This implies that we would expect to see phenotypic delay in the evolution of resistance to some antibiotics, but not to others. In particular, we would expect phenotypic delay due to dilution if the antibiotic acts by binding to its molecular target to make a toxic adduct, and resistance involves production of a resistant target. This is the case for fluoroquinolone antibiotics, which bind to DNA gyrase, causing DNA double-strand breaks (Table 1); resistance is caused by production of mutant gyrase with lower affinity to the antibiotic [83]. The fact that both Boe et al. [47] and Lee et al. [46] observed discrepancies in fluctuation test data for resistance to the fluoroquinolone nalidixic acid is consistent with this expectation.

## Assumptions of the model

Our simulations and theoretical calculations have involved a number of simplifying assumptions. Firstly, we ignore any possible fitness costs of mutations, assuming equal growth rates for wild-type and mutant cells in the absence of the antibiotic. While resistance mutations can incur a fitness cost [84, 85], many clinically-relevant mutations have either no cost or even provide a small growth advantage [85, 86].

For the molecular dilution mechanism, we have assumed that the degradation rate of target molecules is negligible, so that sensitive molecules can only be removed through cell division and dilution. While this seems to be (mostly) the case for bacterial enzymes targeted by antibiotics [87, 88], it may not be true for mammalian cells in which degradation plays a bigger role than dilution [89].

We have also assumed here that in the dilution mechanism, *all* sensitive molecules need to be removed for the cell to become phenotypically resistant, and that each cell has initially the same number of sensitive molecules. In reality, resistance is likely to gradually increase as the number of sensitive molecules decreases, and the total number of target molecules may vary among different cells. Our general conclusions remain valid in this case, but the mutant distribution may change. To construct more accurate models, we need measurements of the degree of antibiotic sensitivity as a function of the intracellular numbers of resistant and sensitive antibiotic targets. While technically challenging, such measurements could be carried out e.g. by fluorescent labelling of target molecules [57]. A starting point for such a detailed model could be to assume the production of sensitive molecules follows the model for protein production of the accumulation mechanism. The value of $n$ per cell would then depend on the number of molecules at the time of mutation, which fluctuates around the mean number of molecules produced per cell division (S1 Text, Section 1.2).

## Experimental tests for phenotypic delay

Sun et al. have demonstrated phenotypic delay by tracking expression of a genetically engineered fluorescent marker in bacterial lineages, and they attributed it to polyploidy [31]. However, their work did not involve *de novo* mutations. Detecting and explaining the mechanism of phenotypic lag due to spontaneous mutations would be much more challenging. Our work suggests that, at least in principle, the mutant number distribution obtained in fluctuation tests could be used to detect the existence of a phenotypic delay caused by molecular dilution, although this would require many replicate experiments. Another possible method could rely on differences in the probability and timing of phenotypic resistance in random lineages. A mother-machine type of experiment in which many lineages can be tracked and exposed to an antibiotic at controlled times could help to determine the contribution of different mechanisms to phenotypic lag. Yet another approach would be an experiment similar to the thought experiment from Fig 1, in which a mutagen such as UV irradiation creates a burst of mutants. Other signatures of phenotypic delay may be detected in experiments where the timing of antibiotic exposure, and of resistance evolution, can be precisely controlled, for example in turbidostat-like continuous culture devices [90].

## Broader significance of phenotypic delay

We have shown here that phenotypic delay (caused by molecular dilution) can affect mutation rate estimates from fluctuation tests, as well as the probability that a bacterial infection survives antibiotic treatment. Phenotypic delay may also affect other processes. For example, it was recently shown that a delay in evolutionary adaptation can lead to coexistence of spatial populations, in cases where immediate adaptation would eradicate coexistence [91, 92]. A delay in

evolutionary adaptation has also been postulated to explain the effect of antibiotic pulses of different lengths on the probability of resistance emerging [93]. Thus, mechanistic understanding of phenotypic delay may be of broad relevance in bacterial evolution.

## Methods and models

In all our simulations we use an agent-based model to simulate how mutated cells gain phenotypic resistance. Each cell has a number of attributes depending on the studied mechanism, such as the numbers of sensitive and mutated DNA copies, and the numbers of sensitive and resistant proteins, as specified below. Cells divide after time $t_d$ since last division.

In our population-level simulations (section *Modelling the emergence of phenotypic delay*), we simulate 100 cells which have just become genetically resistant. Population-level simulations are repeated 1,000 times and single-cell simulations are repeated 10,000 times.

### Modelling effective polyploidy

To describe how the copy number (ploidy) $c$ changes during cell growth and division we use the Cooper-Helmstetter model [32]. We assume that it takes $t_1 = 40$ min for a DNA replication fork to travel from the origin of replication to the replication terminus, and that the cell divides $t_2 = 20$ min after DNA replication termination ($t_1 = C$ and $t_2 = D$ in the original nomenclature of Ref. [32]; values representative for *Escherichia coli* strain B/r). During balanced ("steady state") growth assumed in this work, the number of chromosomes must double during the time $t_d$ between cell divisions (population doubling time). This means that for any $t_d < t_1 + t_2 = 60$ min, the cell must have multiple replication forks and more than one copy of the chromosome. The number of chromosomes will change during cell growth: it will double some time before division, and halve just after the division. If $t_{\text{ini}}$ is the time, since the last division, at which new replication forks are initiated, we must have $((t_{\text{ini}} + t_1) \bmod t_d = t_d - t_2)$. This equation states that the time when a replication round, initiated in the parent cell, finishes in the offspring cell ($(t_{\text{ini}} + t_1) \bmod t_d$) must be the same as the time $t_d - t_2$ when the cell division process (lasting $t_2$ min) is initiated. It can be shown that this gives ($t_{\text{ini}} = t_d - (t_1 + t_2) \bmod t_d$). We proceed in a similar way to determine the time $t_{\text{rep}}$ at which a gene which confers resistance is replicated. If the gene is located in the middle of the genome, as is the case for the *gyrA* gene relevant for fluoroquinolone resistance, it will be copied $t_1/2$ minutes after chromosome replication initiation. This implies that

$$t_{\text{rep}} = t_d - \left( \left( t_2 + \frac{t_1}{2} \right) \bmod t_d \right). \tag{2}$$

At this time point during the cell cycle the copy number of the gene of interest will double. The effective polyploidy immediately after this event is maximal and equal to

$$c = 2^{\lceil \frac{t_1/2 + t_2}{t_d} \rceil}, \tag{3}$$

where $\lceil \ldots \rceil$ denotes the ceiling function. We use $c$ from Eq (3) as the control variable in simulations of the polyploidy model.

To simulate a cell or a population of cells with effective polyploidy we use the following algorithm. We initialize the simulation with all cells having $c/2$ sensitive alleles. Cells replicate in discrete generations every $t_d$ minutes. The number of allele copies doubles at $t_{\text{rep}}$ (Eq (2)) since the last division in such a way that a sensitive/resistant allele gives rise to a sensitive/resistant copy, respectively. Sensitive alleles have a probability $\mu$ of mutating to a resistant allele. When a cell divides, the copies are split between the two daughter cells, with those linked by

the most recent replication fork ending up in the same cell. We assume that the resistance mutation is recessive, which implies that a cell becomes resistant when all of its gene copies are resistant.

## Modelling the dilution of sensitive molecules

For the dilution mechanism, we track the number of sensitive target molecules in each cell. We assume that at time zero, all cells have $n$ sensitive target molecules and no resistant ones. When a mutation happens, we suppose that the mutated cell begins to produce resistant target molecules and ceases to produce new sensitive molecules. At cell division, the sensitive molecules are partitioned between the two daughter cells following a binomial distribution with probability 0.5. We consider that a cell becomes phenotypically resistant when it contains no sensitive molecules. In S1 Text we relax this assumption and study the case where a cell is considered resistant when the number of sensitive molecules falls below a (non-zero) threshold value (S5 Fig).

## Modelling the accumulation of resistance-enhancing molecules

To model the accumulation of resistance-enhancing molecules, we explicitly simulate the production of $M_p$ resistance-enhancing molecules per cell cycle and their stochastic division between daughter cells, via a binomial distribution, at cell division. A cell is considered resistant when it contains more than $M_r$ resistance-enhancing molecules. In all simulations we fix $M_r = 1000$ and vary $M_p$ to explore a range of $m = \frac{M_p}{M_r}$ between 0.1 and 2.

## Combining effective polyploidy and molecular dilution

To include both effective polyploidy and molecular dilution, we track explicitly the total gene copies, the resistant gene copies and the number of sensitive proteins, as explained in Secs. *Modelling effective polyploidy* and *Modelling the dilution of sensitive molecules*. We assume that the number of resistant proteins produced in one cell cycle is proportional to the ratio of resistant to total gene copies. Both types of proteins (sensitive and resistant) are partitioned at cell division following a binomial distribution with probability 0.5. We consider that a cell becomes phenotypically resistant when it contains no sensitive molecules.

## Simulating a growing infection

We start our simulations with 100 sensitive bacteria. Bacteria reproduce in discrete generations with doubling time $t_d$. Upon reproduction, each bacterium can mutate with probability $\mu = 10^{-7}$. When the population reaches $10^7$ cells, all phenotypically sensitive cells are removed (killed); this represent antimicrobial therapy. We repeat the simulation 1000 times to obtain the survival probability as a fraction of simulations in which phenotypically resistant cells emerge before the population dies out.

## Simulating Luria-Delbrück fluctuation tests

To generate mutant size distributions for realistically large population sizes of sensitive cells required for comparing the model with experimental data, we use an algorithm based on Çinlar's method [94, 95]. The algorithm does not simulate the sensitive population explicitly, but it generates a set of times $\{t_i\}$ at which mutants emerge from the exponentially growing sensitive population:

**Algorithm 1:**

```
1 Initialize t = 0, s = 0 tᵢ = [];
2 while t ≤ t_f do
3   s ← s - log(U(0, 1));
4   t ← (1/λₛ)log((1-μ)/(μNᵢ) s + 1);
5   tᵢ.extend(t)
6 end
7 return tᵢ
```

Here $t_f = \frac{\ln(N_f/N_i)}{\lambda_s}$ is the final time, $N_f$ is the final population size, $N_i$ is the initial population size, $\lambda_s = \frac{\ln(2)}{t_d}(1 - \mu)$ is the growth rate of the sensitive bacteria, $t_d$ is the doubling time, $\mu$ is the mutation probability and $U(0, 1)$ is a random variable uniformly distributed between 0 and 1. Formally, $\{t_i\}$ are the times generated from a Poisson process over the interval $[0, t_f]$ with rate $\left(\mu\lambda_s e^{\lambda_s(1-\mu)\tau}\right)_{0\leq\tau\leq t_f}$.

For each $t_i$, we then calculate the number of generations until the final time $t_f$ as

$$g_i = \frac{t_f - t_i}{t_d}. \tag{4}$$

For all of the simulations in sections *Phenotypic delay due to dilution changes the Luria-Delbrück distribution and biases mutation rate estimates* and *Mutant number distributions may support the existence of phenotypic delay*, we assume $t_d = 60$ min. We then simulate each clone for $g_i$ generations, including dilution of sensitive target molecules, and measure the number of resistant cells for each clone. Finally, we measure the number of resistant cells for each replicate by summing up over all clones.

## Approximate Bayesian computation

We use an approximate Bayesian computation method to determine the posterior probabilities of the non-delay and the dilution model. Briefly, the method relies on generating many (here: $10^4$) independent samples of the simulated experiment mimicking Boe et al. [47] for both models. Model parameters are sampled from suitable prior distributions, we then select samples that approximate well the real data, and calculate the fraction of best-fit samples corresponding to each model.

A single sample corresponds to 1104 simulated replicates of the fluctuation experiment at fixed parameters, for a given model. For each sample, parameters are randomly chosen from the following prior distributions: $\log_{10}(\mu)$ uniform on $[-10, -8]$, and $\log_2(n)$ uniform on $[0, 8]$ (for the delay model). The tail cumulative mutation function

$$F(k) = \text{Number of experiments yielding} \geq k \text{ mutants}, \quad 0 \leq k \leq 513, \tag{5}$$

is calculated for each sample $i$ ($F_i$), and also for the experimental data from Boe et al. [47] ($F_{\text{obs}}$). $F$ is undefined for $k \geq 514$ as the authors of [47] grouped replicates yielding more than 512 mutants. We then select 100 out of the $2 \times 10^4$ ($10^4$ from each model) generated samples with the smallest Euclidean distance $||F_i - F_{\text{obs}}||_2$ (simulated distributions closest to the experimental data). The proportion of these which come from the phenotypic delay model is an approximation of the posterior probability that the experimental data was generated by the delay model (under the assumption that the experimental data was generated by one of the models). In reality the data generation process is likely to be far more complex than our idealised models, but the posterior probability of 0.97 implies the delay model provides a superior explanation compared with the model with no delay.

To examine the validity of our approach, we performed cross validation. For each model we randomly chose one sample corresponding to that model. We then computed the probability the simulated data was generated by the model with phenotypic delay, via the approximate method detailed above. This was carried out 500 times for each model. The proportion of simulations that were misclassified (as being with delay when they were not, or vice versa) was low (0.007, Fig 5c), showing that our model selection framework is able to discriminate between the two models. We provide a further sensitivity analysis of this inference method in S1 Text, Section 7.

## Supporting information

**S1 Text. Supplementary information.** Mathematical derivations, additional model variants, sensitivity analysis.
(PDF)

**S1 Table. Numerical values used to generate all graphs.** An Excel spreadsheet with multiple tabs, each corresponding to a single figure.
(XLSX)

**S1 Fig. Single-lineage probability of developing resistance.** (a) We follow a single bacterium which has just mutated and has the resistant allele in one of its chromosomes. When it divides, we choose one of the two daughter cells at random. After a few generations, this cell can become phenotypically resistant. (b) The probability of the cell being resistant as a function of the number of generations from the genetic mutation for the dilution mechanism (dots: simulation, lines: theory Eq (S1)). (c) Same as (b) for the effective polyploidy mechanism. (d) Same as (b) for the accumulation mechanism (only simulations).
(TIF)

**S2 Fig. Expected number of generations until a phenotypically resistant cell emerges.** We start with $x = 100$ cells that just mutated, and repeat the simulation 500 times for each data point. "Analytic approximation" refers to Eq (S6).
(TIF)

**S3 Fig. Biasing the segregation of sensitive molecules at division leads to a decrease in the phenotypic delay both at the (a) single-cell and (b) population level.** Blue curve represents an unbiased case ($p = 0.5$), orange curves is the biased case ($p = 0.62$). In all cases, $n = 1000$.
(TIF)

**S4 Fig. Effect of dependence of the number of target molecules on the doubling time $t_d$ for the combined model.** (a) Probability of resistance as a function of time (generations) for different doubling times (determined by ploidy $c$) when the number of target molecules $n$ depends on $t_d$. (b) Same as (a) but for the model in which $n$ does not depend on $t_d$. (c) Probability of survival for a simulated infection (see section 2.3 and Fig 3 in the main text) for a combined model when the number of target molecules depends on the growth rate. (d) Same as (c) but for the model in which $n$ does not depend on $t_d$.
(TIF)

**S5 Fig. A partial dilution mechanism decreases the phenotypic delay.** (a) Single-cell and (b) population level simulated experiments as a function of $n_r$, the number of sensitive molecules allowed for resistance to emerge. In all cases, the total number of molecules $n = 1000$.
(TIF)

**S6 Fig. Maximum likelihood estimates of $\mu$ from 1000 simulations mimicking the experiment of Ref. [46] with known $\mu = 3.98 \times 10^{-9}$ (black vertical line) for the no-delay model.** The mutation probability can be underestimated by a factor of 2 (95% of simulations yielded estimates between red vertical lines), whereas Ref. [46] reports a factor of 9.5 difference between $\mu$ obtained from DNA sequencing and fluctuation tests. The Lee et al. result [46] cannot be thus explained by the no-delay model.
(TIF)

**S7 Fig. Sensitivity analysis for model selection.** (a) The probability of the Boe et al. data [47] coming from the delay model as a function of the number of simulation runs. The runs were randomly sampled from the original bank of simulations and the probability of the delay model was estimated. The process was repeated 10 times. Error bars are the maximum and minimum probability estimated, with the centred dot as the mean. (b) The probability estimate for the probability of the data [47] coming from the delay model as a function of $N_{\text{thresh}}$.
(TIF)

**S8 Fig. Full distribution for the Luria-Delbrück simulations of the dilution model presented in Fig 4 in the main text.** a) Distributions for both models for a fixed $\mu = 10^{-7}$. (b) Distributions for the case when $\mu$ in the dilution model has been adjusted to minimize the difference to the no-delay model.
(TIF)

## Acknowledgments

We thank Helen Alexander (University of Edinburgh) for helpful discussions.

## Author Contributions

**Conceptualization:** Martín Carballo-Pacheco, Michael D. Nicholson, Elin E. Lilja, Rosalind J. Allen, Bartlomiej Waclaw.

**Data curation:** Martín Carballo-Pacheco, Michael D. Nicholson.

**Formal analysis:** Martín Carballo-Pacheco, Michael D. Nicholson.

**Funding acquisition:** Rosalind J. Allen, Bartlomiej Waclaw.

**Investigation:** Martín Carballo-Pacheco, Michael D. Nicholson.

**Methodology:** Martín Carballo-Pacheco, Michael D. Nicholson, Bartlomiej Waclaw.

**Project administration:** Rosalind J. Allen, Bartlomiej Waclaw.

**Resources:** Rosalind J. Allen, Bartlomiej Waclaw.

**Software:** Martín Carballo-Pacheco, Michael D. Nicholson.

**Supervision:** Rosalind J. Allen, Bartlomiej Waclaw.

**Validation:** Martín Carballo-Pacheco, Michael D. Nicholson.

**Visualization:** Martín Carballo-Pacheco, Michael D. Nicholson.

**Writing – original draft:** Martín Carballo-Pacheco, Michael D. Nicholson.

**Writing – review & editing:** Martín Carballo-Pacheco, Michael D. Nicholson, Elin E. Lilja, Rosalind J. Allen, Bartlomiej Waclaw.

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
