## [Decision Letter · Decision Letter 0]

27 Mar 2020

Dear Waclaw,

Thank you very much for submitting your manuscript "Phenotypic delay in the evolution of bacterial antibiotic resistance: mechanistic models and their implications" for consideration at PLOS Computational Biology. As with all papers reviewed by the journal, your manuscript was reviewed by members of the editorial board and by several independent reviewers. The reviewers appreciated the attention to an important topic. Based on the reviews, we are likely to accept this manuscript for publication, providing that you modify the manuscript according to the review recommendations.

Sincerely,

Mark M. Tanaka

Associate Editor

PLOS Computational Biology

Stefano Allesina

Deputy Editor

PLOS Computational Biology

[LINK]

Reviewer's Responses to Questions

**Comments to the Authors:**

Reviewer #1: Attached.

Reviewer #2: This paper discusses phenotypic delay as an overlooked part of bacterial antibiotic resistance evolution with important implications for the experimental estimation of mutation rates and the survival in the face of an antibiotic attack. In my opinion, the paper is well-written, and the results are interesting, important, and sound. I have only minor comments that may help to further improve the readability of the paper and provide interesting additional cases.

1. The authors discuss two ways of looking at the problem, per lineage and per population. It is not always entirely clear what exactly the differences are between those two, why they sometimes give different results, and which viewpoint is taken in a given part of the text or figure. For instance, Fig. 1 is concerned with a population where mutations has been introduced, and what is measured is the probability that phenotypic resistance will emerge in that population by some generation post-mutation (it may be helpful to include "post-mutation" to the axis label to reduce confusion that mutation rates are at all involved here). However, the explanation in the text refers to lineages quite frequently, which may be confusing to the readers.

2. In section 2.3 and 2.4, the authors say "this is due to the cancellation of two effects [...]". While the explanation makes intuitive sense, it may improve intuition further to show the expression and the cancellation explicitly to make this point.

3. In some cases of phenotypic delay, the evolutionary dynamics depends on the growth rate. It would be interesting to briefly discuss two more cases: how does individual growth rate variation impact the dynamics, e.g., if the population houses a fraction of slow-growing cells that not only typically have a higher probability of surviving an antibiotic attack by, e.g., betalactams, but that also, per Fig. 3e, have a higher probability of survival due to phenotypic delay. And secondly, how does the dynamics change for the important case of sub-lethal antibiotic concentrations, where growth rates are decreased but non-zero?

4. For Fig. 4, it would have loved to see, perhaps as a SI figure, the full distribution on a log-log scale. Does the typical 1/x scaling of the cumulative distribution change, if so, in which regimes and can the scaling be identified? This is roughly shown in Fig. 5b, but a more thorough discussion would be appreciated. Is the staircase-like shape of the simulation line in Fig. 5b a consequence of the non-overlapping generations? This should be explained in the caption.

**Have all data underlying the figures and results presented in the manuscript been provided?**

Reviewer #1: Yes

Reviewer #2: No: If there is a link to data and code repositories I did not find it in the manuscript.

PLOS authors have the option to publish the peer review history of their article (what does this mean?). If published, this will include your full peer review and any attached files.

Reviewer #1: No

Reviewer #2: Yes: Matti Gralka
---

## [Editor Report · Decision Letter 1]

6 May 2020

Dear Waclaw,

We are pleased to inform you that your manuscript 'Phenotypic delay in the evolution of bacterial antibiotic resistance: mechanistic models and their implications' has been provisionally accepted for publication in PLOS Computational Biology.

Best regards,

Mark M. Tanaka

Associate Editor

PLOS Computational Biology

Stefano Allesina

Deputy Editor

PLOS Computational Biology

We are satisfied that you have addressed the reviewers' comments.

---

## [Editor Report · Acceptance letter]

22 May 2020

PCOMPBIOL-D-20-00158R1 

Phenotypic delay in the evolution of bacterial antibiotic resistance: mechanistic models and their implications

Dear Dr Waclaw,

I am pleased to inform you that your manuscript has been formally accepted for publication in PLOS Computational Biology. Your manuscript is now with our production department and you will be notified of the publication date in due course.

With kind regards,

Laura Mallard
